**Introduction**

Wu et al in this paper explore an adaptive computation method that can be efficiently applied to an existing generative ODQA model. They find that, by replacing the encoder of generative ODQA models with their proposed adaptive passage encoder, they can train an effective adaptive computation policy without tuning the base model. This allows applying adaptive computation to large state-of-the-art generative models, which was previously challenging computation wise. Their experimental results show that their method produces more accurate results than a state-of-the-art generative model on both NaturalQuestions and TriviaQA, and it outperforms the previous AC method by a large margin. We want to reproduce some results of their expriments.

**Scope of Reproducibility**

In this work we want to reproduce this paper (original paper) result. This paper proposed a APE-FiD-base model and compare its result with another paper (FiD-base model) . Our goal is to examine these expriments. You can see our implementations in our github repository. All details described in README file completely and you can study it.

**Methodology**

In this work, We used the author codes proposed in this github repository.We used google colab environment and our dedicated configurations was:

- Run Type: GPU
- Disk: 80 GB
- RAM: 12 GB

Due to lack of memory resources, we had to run the experiments on 1/8 of the original data, and this required making changes to the original code, which is presented in detail in our github repository. Each of our experiments lasted an hour and a half (Totally three hours).

**Results**

We produced almost 50% of the results of the original paper. The results we obtained were lower than the results of the original paper (due to memory constraints we had to experiment on a small portion of the data) and therefore we expected the results to be different from the results of the original paper.

Of course, our results showed the superiority of the proposed method of the original paper, and we observed in our experiments this superiority of the proposed method in the test set data (see our github repository).

**What was easy**

Model training was easy. The input parameters required for model training were well mentioned and also good error messages make error handling feasible. (Training of APE-FiD base model was easier  train than the FiD-base model)

**What was difficult**

One of our main problems was that we did not have access to the original paper data and we encountered a 403 error while downloading and we had to use the FiD-base data.

Also, data preprocessing was hard task for us. Due to the large volume of data we had to split the data and this splitting had to be done with the indexes in mind because the different parts of the data connected to each other and we had to pay attention to the original data format. Of course, we took this into account and preprocessed our data with respect to this consideration.

**Communication with original authors**

We did not have any communication with original authors and just used their github repositories.