# OpenReview forum: "Training Adaptive Computation for Open-Domain Question Answering with Computational Constraints"
_ML_Reproducibility_Challenge/2021/Fall — Reject_

### Official Review · Reviewer_PYxS · 2022-02-28
**The report is not very self-contained and potentially violates the double-blind rule**

**Rating:** 3
**Confidence:** 4

**Review:**

The paper looks into reproducing the results in " Training Adaptive Computation for Open-Domain Question Answering with Computational Constraints" by Wu et al. from ACL2021.

Strengths
       -  The paper is able to provide some insights on some of the challenges of reproducing the original work.

Weaknesses
	- The reproducibility report is not very self-contained.
        - The paper may have violated the double-blind rule.
	- Some details about the evaluation methodology and results are not well explained.

Comments:

The reproducibility report has several issues.

First, it is not very self-contained. Hyperlinks are used to refer to the original paper and baselines, instead of using proper citations. The reviewer has to constantly refer to the original paper to figure out what are the claims the original paper makes and what this report tries to verify. The result section is also quite simplified without much analysis.

A github repo link is provided as well as a hyperlink, which appears to have more detailed instructions on how the results are reproduced.  The paper says "all details described in README file completely and you can study it", but the external link is not an anonymous github URL and is under a public user's repo. This violates the double-blind rule here at "https://paperswithcode.com/rc2021/faq", which states that code link should be anonymous.

Some of the evaluation details are missing. The paper says it reproduces 50% of the results of the original paper, but it does not explicitly say which 50%. It does not provide a rationale on how that 50% are chosen. Also, the paper says the obtained results are lower than the results of the original paper and claims that it is caused by memory constraints. However, this again is not supported by detailed analysis, e.g., what is the memory consumption, why using a smaller portion of the data would reduce the memory consumption given that the memory usage is primarily decided by the model states and batch sizes.

Finally, the paper mentions that data preprocessing is a big challenge for reproducing the results. However, it only provides very high-level descriptions on how it resolves this issue, e.g., by splitting the data with the indexes in mind.  It would be better if the authors could explain what the original data format is and how exactly this data split processing handles that data format.

---

### Official Review · Reviewer_VrAK · 2022-03-01

**Rating:** 4
**Confidence:** 3

**Review:**

Reviewing this submission was impractical as the submitted element was the reproducibility summary. However within this summary some of the difficulties were clearly illustrated. That being said the precise experiments that were attempted to be reproduced were unclear.

---

### Official Review · Reviewer_HQon · 2022-03-28
**Lack Results**

**Rating:** 3
**Confidence:** 5

**Review:**

The report described an attempt to reproduce the paper of Wu et al 2021, which is an idea of using an adaptive passage encoder for open-domain QA models. The report describes the codebase used, which is from the original paper authors, and the experimental environment, which is on Google colab. The report mentioned the difficulty of reproducing results using the full data from the original paper due to resource constraints, and the lack of access for the original paper data.

In spite of the technical details given by the report, it lacks any numerical results that show how the results look. Even if we are using a subset of the datasets to reproduce results, the numbers could still be useful as a complement to the original paper. Missing this, and the lack of any analysis thereof, are the main reasons for the rejection recommendation.

---

### Meta-Review · Area_Chair_6QeA · 2022-04-09

**Recommendation:** Reject
**Confidence:** 5

**Metareview:**

This report seems incomplete, it only presents the first page of the report (not the body).

---

### Decision · Program_Chairs · 2022-04-09

Reject